# Reduced purine biosynthesis in humans after their divergence from Neandertals

Vita Stepanova[1,2†*], Kaja Ewa Moczulska[3†‡], Guido N Vacano[4†], Ilia Kurochkin[1†], Xiangchun Ju[3,5†], Stephan Riesenberg[3], Dominik Macak[3], Tomislav Maricic[3], Linda Dombrowski[3], Maria Schörnig[3], Konstantinos Anastassiadis[6], Oliver Baker[6§], Ronald Naumann[7], Ekaterina Khrameeva[1], Anna Vanushkina[1], Elena Stekolshchikova[1], Alina Egorova[1], Anna Tkachev[1], Randall Mazzarino[4#], Nathan Duval[4], Dmitri Zubkov[1], Patrick Giavalisco[8], Terry G Wilkinson[4], David Patterson[4*], Philipp Khaitovich[1*], Svante Pääbo[3,5*]

[1]Skolkovo Institute for Science and Technology, Skolkovo, Russian Federation; [2]Institute for Information Transmission Problems, Russian Academy of Sciences, Moscow, Russian Federation; [3]Max Planck Institute for Evolutionary Anthropology, Leipzig, Germany; [4]The Eleanor Roosevelt Institute and Knoebel Institute for Healthy Aging, University of Denver, Denver, United States; [5]Okinawa Institute of Science and Technology, Onna-son, Japan; [6]Center for Molecular and Cellular Bioengineering, Biotechnology Center, Technical University Dresden, Dresden, Germany; [7]Max Planck Institute of Molecular Cell Biology and Genetics, Dresden, Germany; [8]Max Planck Institute for Biology of Ageing, Cologne, Germany

*For correspondence:
vita.stepanova@skolkovotech.ru (VS);
david.patterson@du.edu (DP);
p.khaitovich@skoltech.ru (PK);
paabo@eva.mpg.de (SP)

†These authors contributed equally to this work

Present address: ‡The Francis Crick Institute, London, United Kingdom; §King's College London, New Hunt's House, Guy's Campus, London, United Kingdom; #Schepens Eye Research Institute of Massachusetts Eye and Ear Infirmary, Department of Ophthalmology, Harvard Medical School, Boston, Massachusetts, United States

Competing interests: The authors declare that no competing interests exist.

**Abstract** We analyze the metabolomes of humans, chimpanzees, and macaques in muscle, kidney and three different regions of the brain. Although several compounds in amino acid metabolism occur at either higher or lower concentrations in humans than in the other primates, metabolites downstream of adenylosuccinate lyase, which catalyzes two reactions in purine synthesis, occur at lower concentrations in humans. This enzyme carries an amino acid substitution that is present in all humans today but absent in Neandertals. By introducing the modern human substitution into the genomes of mice, as well as the ancestral, Neandertal-like substitution into the genomes of human cells, we show that this amino acid substitution contributes to much or all of the reduction of de novo synthesis of purines in humans.

## Introduction

Modern humans differ dramatically from their closest evolutionary relatives in a number of ways. Most strikingly, they have developed rapidly changing and complex cultures that have allowed them to become more numerous and widespread than any other primates and closely related hominins such as Neandertals. This unique historical development has at least to some extent biological roots. However, although large numbers of traits have been identified as being unique to humans or been suggested to be so (e.g. *Tomasello, 2019*, *Varki and Altheide, 2005*), it has proven difficult to identify the genetic and biological underpinnings of such traits. One reason may be that many of them are genetically complex (*Phillips, 2008*) and represent differences in degree rather than absolute differences between humans and other groups. In particular, it is unclear if any single nucleotide substitution or other genomic change that occurs in all or almost all humans today, but is not seen among the closest evolutionary relatives of present-day humans, the Neandertals, and Denisovans (*Pääbo, 2014*), has biological consequences. A duplicative transposition in modern humans of the gene *BOLA2* which is involved in iron metabolism has been described (*Nuttle et al., 2016*) in a

region associated with autism but the functional consequences are unknown. A recent study introduced a modern human-specific amino acid substitution in the splice factor NOVA1 and suggested that it has drastic effects on the differentiation of brain organoids (*Trujillo et al., 2021*), an observation that warrants further study.

To identify traits that may have comparatively simple genetic architectures yet influence the biology of modern humans, we focus here on metabolic differences. By analyzing the metabolomes of muscle, kidney and three different regions of the brain from humans, chimpanzees and macaques, we find that many aspects of amino acid metabolism differ between humans and the other two primates in all tissues analyzed. Among metabolic pathways, purine biosynthesis is less active in several human tissues including the brain. In purine biosynthesis, we find that metabolites downstream of the enzyme adenylosuccinate lyase [ADSL (EC 4.3.2.2)] occur at lower concentrations in humans than in the other primates. ADSL carries an amino acid substitution that is unique to modern humans relative to apes and Neandertals and Denisovans (*Castellano et al., 2014*) and has been shown to affect the stability of the enzyme (*Van Laer et al., 2018*). By introducing the modern human-like substitution in the genome of mice, and the ancestral, Neandertal-like substitution in the genomes of human cells, we show that this substitution contributes to much or all of this metabolic change in present-day humans.

## Results

### Metabolite changes unique to humans

To identify metabolites that have changed their concentration in humans relative to monkeys and apes, we measured compound concentrations in prefrontal cortex, primary visual cortex, cerebellum, skeletal muscle and kidney in four humans, four chimpanzees, and four macaques using mass spectrometry coupled with capillary electrophoresis (CE-MS), a technique suitable for detection of small hydrophilic compounds (*Figure 1a,b*). The number of metabolites annotated in the five tissues varied between 166 and 197, 160 and 209, and 145 and 192 in the three species, respectively (*Supplementary file 1*).

In each tissue, we identified metabolites that did not differ significantly with respect to their mass spectrometric peak intensities between the macaques and the chimpanzees, but differed significantly, and in the same direction, between humans and macaques as well as between humans and chimpanzees. Whereas we find no such metabolites in skeletal muscle and kidney, we find metabolites peak intensities and thus concentrations differ in humans relative to the other two primates in two of the three parts of the brain analyzed.

In cerebellum, 22 metabolites have higher concentrations in humans whereas no metabolites have lower concentrations (*Supplementary file 2*). Eighteen of the 22 metabolites are amino acids. In prefrontal cortex, we detected five metabolites with lower concentrations in humans and none with higher concentrations (*Supplementary file 3*). Three of the five metabolites are purines (inosine monophosphate [IMP], guanosine monophosphate [GMP], adenosine monophosphate [AMP]) and the other two are $NAD^+$ and UDP-N-acetylglucosamine.

### Metabolic pathways

To identify metabolic pathways that may be more or less active in humans than in other primates, we linked metabolites with higher or lower concentrations in humans compared to both chimpanzees and macaques (even if not significantly so) to genes and metabolic pathways using the Kyoto Encyclopedia of Genes and Genomes (KEGG). *Supplementary files 4* and *5* give pathways that contain more genes than expected (p<0.01) that are associated with metabolites present in either higher or lower concentrations in humans in each of the five tissues.

Between 8 and 12 pathways are associated with higher metabolite concentrations in the five human tissues (*Supplementary file 4*). All eight pathways identified in the prefrontal cortex involve amino acid metabolism, eight out of nine pathways in the visual cortex and nine out of 10 in the cerebellum also involve amino acids. In muscle and kidney, 7 and 5 of the 11 and 12 pathways identified, respectively, involve amino acid metabolism. Thus, several aspects of amino acid metabolism seem to be increased in humans relative to other primates.

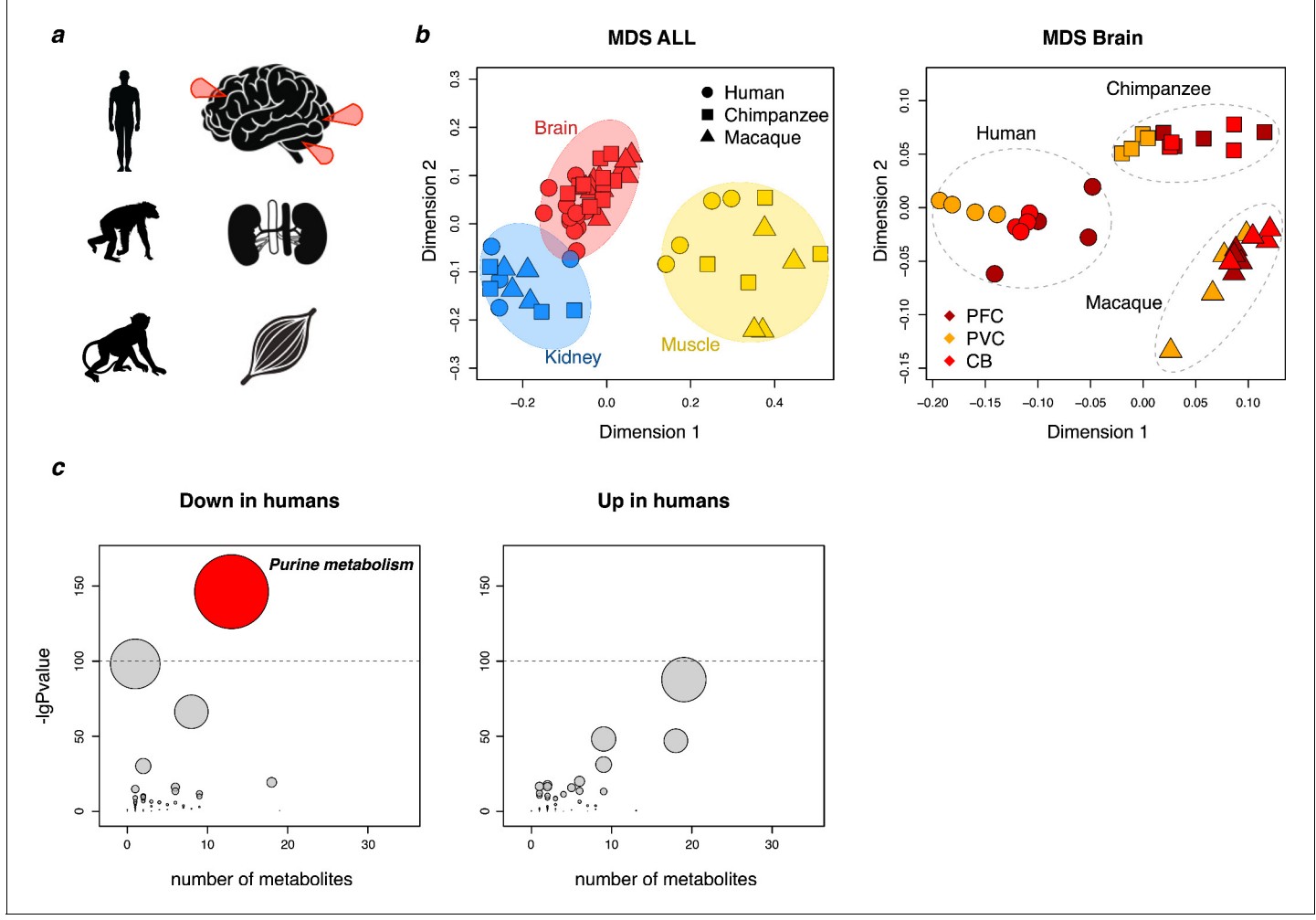

**Figure 1.** Primate metabolome analysis. (**a**) Sketch depicting species and tissues used for metabolite measurements. Red arrows indicate sampled brain regions: the dorsolateral prefrontal cortex, primary visual cortex, and cerebellar cortex. (**b**) Multidimensional scaling (MDS) analysis of all samples (left) and brain samples (right) based on the intensities of all detected metabolic peaks. (**c**) An analysis of KEGG-based metabolic pathway showing for each pathway (circle) the number of detected metabolites (X axis) and the cumulative p-value of the difference (Y axis) between human and non-human primate (chimpanzee and macaque) intensity levels within the pathway in the five tissues. Left: an analysis based on metabolites with lower intensity in humans compared to non-human primates. The oxidative phosphorylation pathway is the second highest. Right: an analysis based on metabolites with higher intensity in humans. The size of the symbols is proportional to the -logarithm-transformed p-value. The dotted line indicates nominal p=10$^{-10}$.

Between 4 and 11 pathways are associated with lower metabolite concentrations in the five tissues (*Supplementary file 5*). Amino acid and peptide metabolism make up one to five of these pathways in the different tissues, suggesting that amino acid metabolism has changed in several ways in humans relative to the other primates. Hence, metabolites in these pathways are present in increased as well as decreased concentrations, often in the same pathways. For example, significant numbers of metabolites in arginine and proline metabolism are present in higher concentrations while other metabolites in the same pathways are present in lower concentrations, sometimes in the same tissues, suggesting that the flux through these pathways has changed.

Metabolites in two pathways are consistently present at lower rather than higher concentrations in humans. One of these pathways is oxidative phosphorylation, which shows decreased metabolite concentrations in the three brain regions. Because oxidative phosphorylation is not affected in muscle and kidneys, it seems that the human brain differs from ape brains in that oxidative phosphorylation in mitochondria is less active. This may deserve future investigation.

The second pathway that stands out is purine biosynthesis where a number of metabolites are present at lower levels in humans (*Figure 1c*). Based on the KEGG definition of the pathway, purine

biosynthesis is decreased in humans relative to apes in the brain as well as in other organs. If we focus on a more restrictive definition of purine biosynthesis (*Marie et al., 2004*), a significant decrease is seen in the three brain regions and not in muscle and kidney (*Figure 2*). Further, three of the five metabolites, IMP, GMP and AMP, which individually have significantly lower concentrations in humans, are products of de novo purine biosynthesis. Taken together, metabolites involved in purines biosynthesis have significantly lower concentrations in humans than in non-human primates in all three brain regions than do metabolites in the other metabolic pathways, (Wilcoxon test,

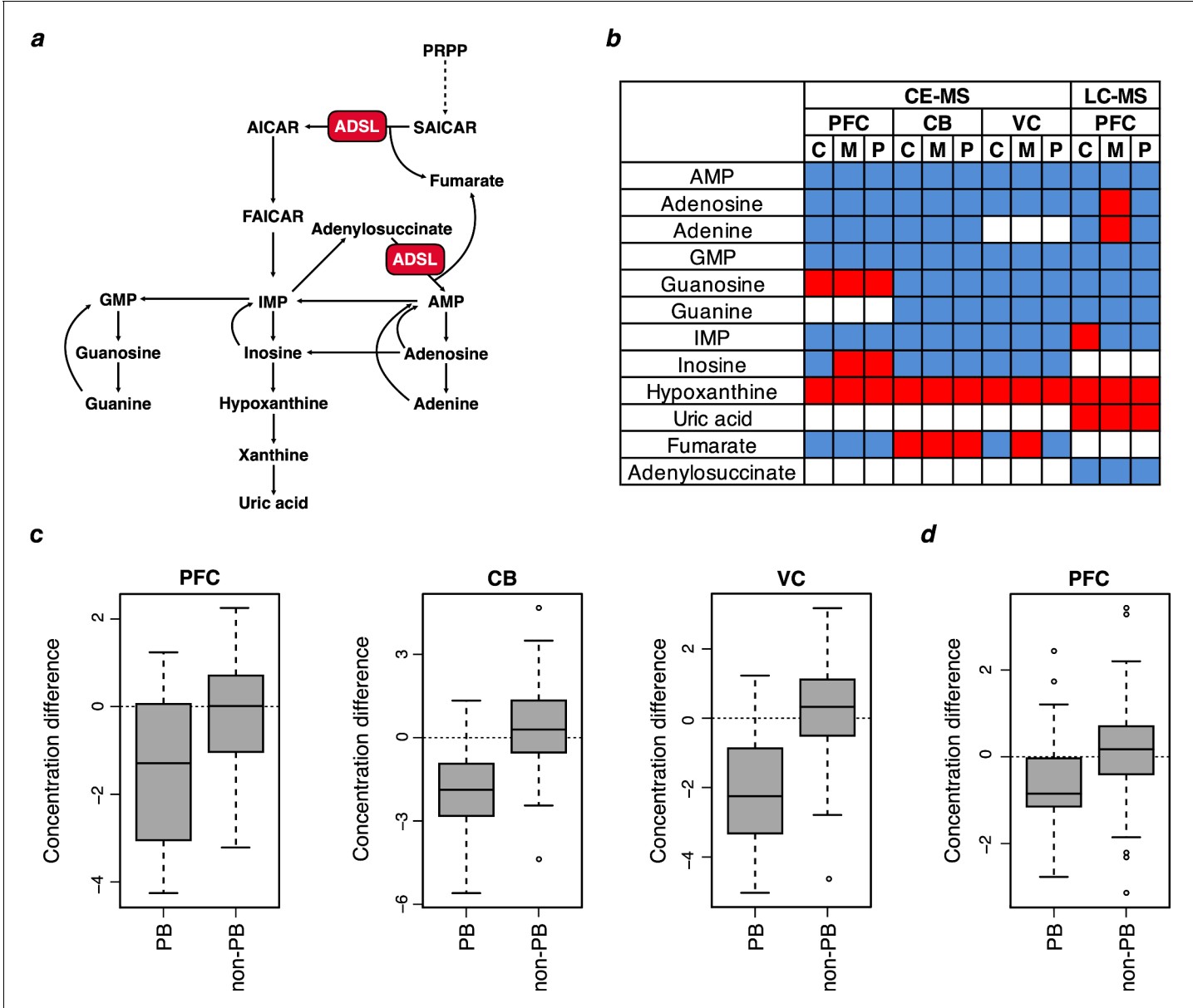

**Figure 2.** Purine metabolite levels in humans relative to chimpanzees and macaques. (**a**) Pathway sketch illustrating the role of ADSL in purine biosynthesis. (**b**) Differences in metabolite levels between humans (n = 4) and chimpanzees (C) (n = 4), macaques (M) (n = 4) and the two non-human primates combined (P) in prefrontal cortex (PFC), cerebellum (CB), primary visual cortex (VC) analyzed by CE-MS. LC-MS measurements in PFC are from *Kurochkin et al., 2019* and involved 40 humans, chimpanzees and macaques, respectively. Red and blue colors indicate higher and lower levels in humans, respectively. c and d. The distributions of metabolite differences between humans and non-human primates for purines biosynthesis (PB) (listed in panel **b**) and other metabolites (non-PB) detected by CE-MS (**c**) and LC-MS (**d**). The Y axis shows fold-change difference between humans and non-human primates. The metabolite concentration values are normalized and log-transformed. Wilcoxon tests for the differences between PB and non-PB distributions: p=0.049 (PFC), p=0.002 (CB), p=0.001 (VC), and p=0.019 (PFC LC-MC).

p=0.049 (PFC), p=0.002 (CB), p=0.001 (VC); *Figure 2c*). Similarly, we find the significantly lower concentrations of purine biosynthesis metabolites in humans compared to chimpanzees and macaques relative to other metabolic pathways when analyzing previously published metabolite data from the PFC of three species (*Kurochkin et al., 2019*) (Wilcoxon test, p=0.019; *Figure 2d*). Thus, purine biosynthesis stands out as down-regulated in humans. This is seen in all tissues but is particularly pronounced in the brain.

## Adenylosuccinate lyase (ADSL) in modern humans

The enzyme adenylosuccinate lyase (ADSL) catalyzes two reactions in de novo purine biosynthesis (*Figure 2a*; *Ciardo et al., 2001*; *Jaeken and Van den Berghe, 1984*). First, it cleaves succinylaminoimidazole carboxamide ribotide (SAICAR) into aminoimidazole carboxamide ribotide (AICAR) and fumarate. Second, it cleaves adenylosuccinate (S-AMP) into adenosine monophosphate (AMP) and fumarate (*Figure 2a*). The compounds AMP, IMP, and GMP that are reduced in the three human brain regions analyzed are situated downstream of ADSL, consistent with the hypothesis that a change in ADSL may have caused the reduction in purine biosynthesis.

ADSL (*Ariyananda et al., 2009*) is a homotetrameric complex where three monomers contribute to each of the four active sites (*Brosius and Colman, 2002*). The gene encoding ADSL in humans is located on chromosome 22q13.1–13.2 and is one of only approximately one hundred protein-coding genes that carry substitutions inferred to change amino acids that are fixed or almost fixed in present-day humans but occur in their ancestral, ape-like state in the genomes of the closest extinct relatives of modern humans, Neandertals and Denisovans (*Pääbo, 2014*). The substitution in *ADSL* in modern humans is an alanine to valine substitution at position 429 (A429V). Neandertals, Denisovans, all other primates and most mammals carry an alanine residue at this position (*Figure 3*). The substitution is located in a protein domain that forms part of the substrate channel over the active site of the enzyme. Amino acid substitutions close to this position cause lowered enzymatic activity and/or lower enzyme stability and result in symptoms that include psychomotor retardation, autism, and epilepsy (*Jurecka et al., 2015*; *Spiegel et al., 2006*; *Zikanova et al., 2010*) as well as alterations in brain structure as seen by magnetic resonance imaging (*Jurecka et al., 2012*). This raises the possibility that the amino acid substitution at position 429 may have changed the activity of ADSL in modern humans after their separation from the lineage leading to Neandertals and Denisovans.

## A mouse humanized for *Adsl*

To investigate how the A429V substitution may affect purine biosynthesis, we introduced this mutation into the *Adsl* gene of a C57BL/6N mouse by injection of the relevant oligonucleotide and CRISPR-*Cas9* into the male pronucleus of fertilized oocytes. Adjacent to the amino acid of interest, at position 428, mice carry an arginine residue whereas primates carry a glutamine residue (*Figure 3*). To avoid possible effects of the rodent-specific arginine residue at position 428 on the function of the amino acid at position 429, we introduced a nucleotide substitution resulting in an R428Q substitution in addition to the V429A substitution in the *Adsl* gene. The two mutations segregate in Mendelian ratios in the mice, and animals heterozygous and homozygous for the two substitutions show no overt phenotypic difference to their wild-type littermates.

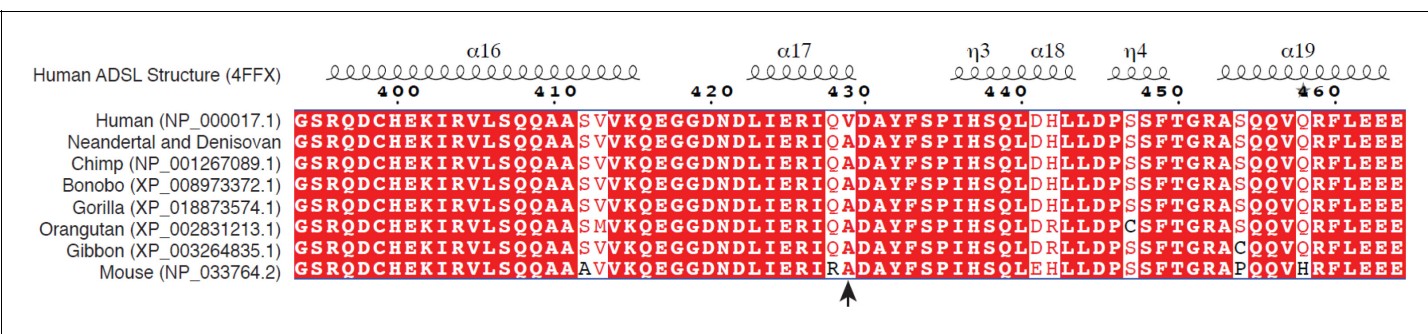

**Figure 3.** ADSL protein sequence. Partial amino acid sequences of ADSL. Accession numbers and amino acid sequences are from NCBI. X-ray structural determination of ADSL (DOI: 10.2210/pdb4FFX/pdb) is from *Ray et al., 2012*. Arrow indicates position 429.

## ADSL activity in the humanized mice

We analyzed the enzymatic activities of ADSL in tissue extracts prepared from mice homozygous for the human-like mutations and their wild-type littermates in cerebellum, cerebral cortex, heart, kidney, liver, lung, muscle, spleen, and testis in 12-week-old mice and (except testis) in 1-week-old mice by measuring the conversion of adenylosuccinate to AMP.

In the wild-type adult mice (n = 10), mean ADSL activity varied from 1.95 nmole/min/μg in liver to 24.4 nmole/min/μg in muscle (*Figure 4b*). In the pups (n = 10), mean enzymatic activity was 5.4% to 57% higher in most tissues and largely paralleled those of the adults (*Figure 4a*). Two exceptions were cerebellum and muscle, where ADSL activity was 41% and 11% lower in the pups than in the adults, respectively.

In the humanized mice, ADSL activity was reduced by 8.7% to 56% in all organs in both adults (n = 11) and pups (n = 12) (*Figure 4*; p<0.05, for all comparisons in both age groups). The relative reduction of the activity in the humanized relative to the wild-type mice is higher in cerebral cortex (43% in pups and 56% in adults) than in the other tissues (8.7%–40%). Thus, young as well as adult mice that carry an ADSL enzyme humanized at positions 428 and 429 have lower ADSL activity in most or all tissues.

## The metabolome of humanized *Adsl* mice

We next analyzed the metabolomes of the nine tissues from 9 to 12 adult homozygous humanized mice and their wild-type littermates by gas chromatography coupled with mass spectrometry (GC-MS). We similarly analyzed eight tissues (no testis) from 10 to 12 one-week-old pups

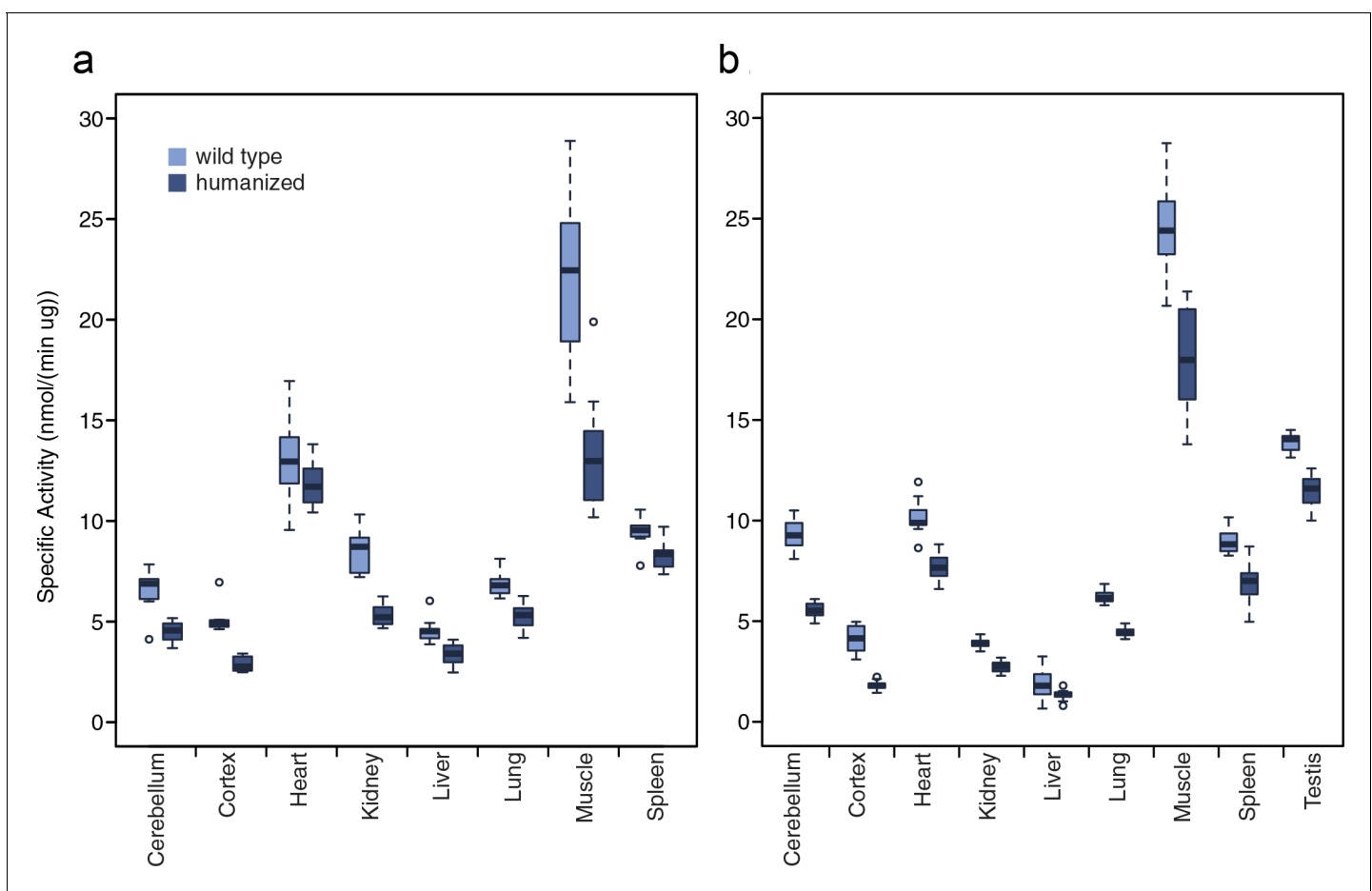

**Figure 4.** ADSL activity in tissues from humanized and wild type mice. (a) One-week-old mice. (b) Twelve-week-old mice. Conversion of adenylosuccinate to AMP was monitored by measuring λ 282 nm absorbance over 20 min (see Materials and methods). Boxes represent upper and lower quartiles, lines extending from the boxes represent the ranges of the data, excluding outliers which are indicated by dots.

(*Supplementary file 6*). The number of putatively identified metabolites detected varied between 176 and 273 in the adult mice and between 310 and 347 in the young mice. Among those, an average of 280 (median = 273) were detected in at least 50% of the adult and young individuals in each of the nine tissues (*Supplementary files 7*, *8*). A principal component analysis using the mass spectrometric intensities of all metabolites detected revealed one to two outlier samples per tissue (*Supplementary file 6*). These were excluded from further analyses. Including these samples in the analyses did not qualitatively affect results.

Among the organs analyzed, only the brain showed significant differences in metabolite concentrations between the wild type and humanized mice (*Figure 5a*). To assess the statistical significance, we use a permutation test based on shuffling of sample labels, which estimates a p-value as the ratio of permutations resulting in an equal or greater number of metabolites with significant concentration differences at a nominal t-test significance cutoff of 0.05. According to this test, 36 metabolites showed significant concentration differences in cerebellum of 12-week-old mice (permutation test, p=0.036) and 45 metabolites showed marginally significant differences in the cerebral cortex of 1-week-old mice (permutation test, p=0.081, *Supplementary file 9*). Thus, the metabolic effects of the two mutations introduced in ADSL are particularly pronounced in the central nervous system.

The concentration differences detected in cerebellum of the 12-week-old mice correlated with differences observed in cerebral cortex, even though differences in cortex did not pass the significance cut-off in the permutation test (Pearson correlation, r = 0.84, p<0.0001, n = 29; *Figure 5—figure supplement 1a*). Similarly, concentration differences detected in cerebral cortex in one-week-old mice correlated with differences observed in cerebellum in the same mice (Pearson correlation, r = 0.77, p<0.0001, n = 45; *Figure 5—figure supplement 1b*). The concentration differences between wild type and humanized mice furthermore correlated between the 1-week-old and 12-

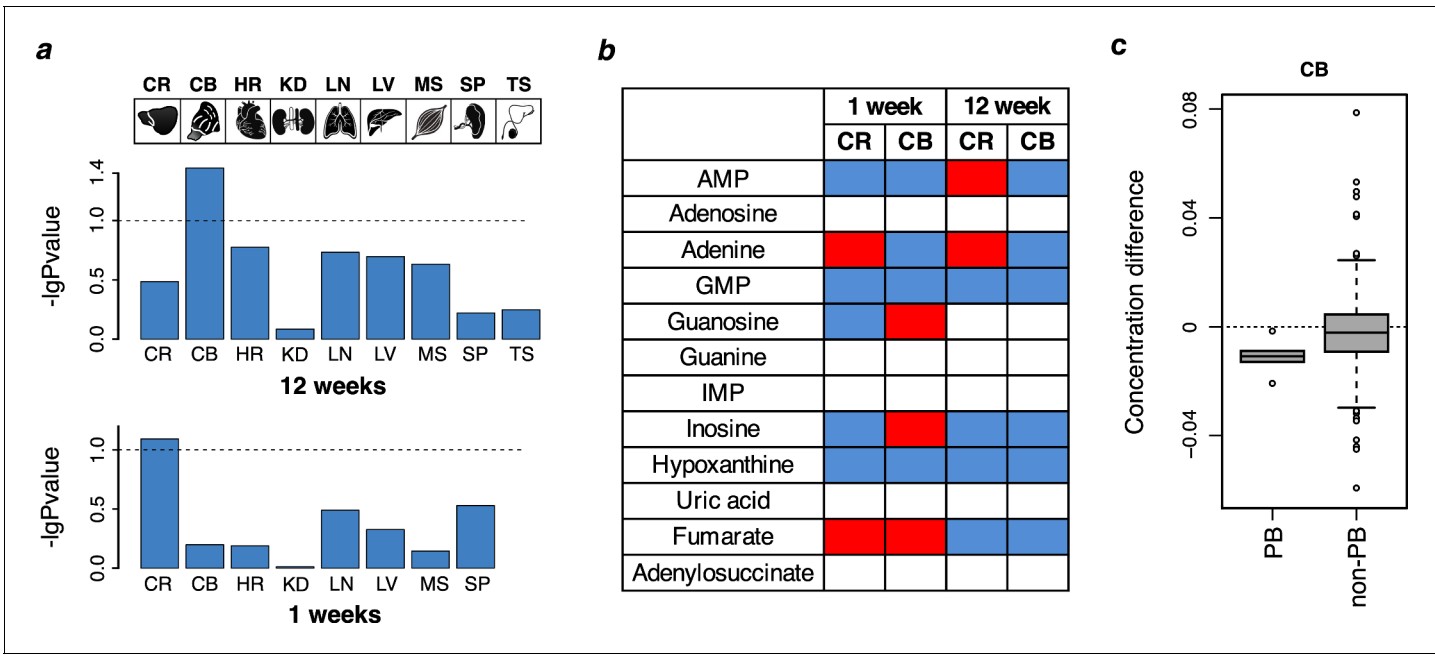

**Figure 5.** Purine biosynthesis in mice 'humanized' for ADSL. (**a**) Data overview displaying -log10 transformed p-values of the metabolite differences between humanized and wild-type one-week (n = 10–12 per group) and 12-weeks-old (n = 9–12 per group) animals based on all detected metabolites (permutation test based on the Student's test output). The tissues analyzed are: brain cortex (CR), cerebellum (CB), heart (HR), kidney (KD), lung (LN), liver (LV), muscle (MS), spline (SP), and testis (TS, 12 weeks only). Dotted lines indicate p=0.1. (**b**) The direction of purine metabolite differences in pairwise comparisons between humanized and wild type mice in the cortex (CR) and cerebellum (CB). Red and blue colors marks higher and lower levels in the humanized mice, respectively, white indicates non-detected metabolites. (**c**) The distributions of metabolite differences between 12-week-old humanized and wild-type mice for purines biosynthesis (PB) (listed in panel **b**) and the other metabolites (non-PB) detected using GC-MS in CB of 12-week-old mice (Wilcoxon test, p=0.04).

The online version of this article includes the following figure supplement(s) for figure 5:

**Figure supplement 1.** Correlations of metabolite differences in mouse tissues.

week-old mice both in cortex and in cerebellum (Pearson correlation, r = 0.49 and r = 0.52, p<0.008, n = 28 and n = 31, respectively; *Figure 5—figure supplement 1e,f*). By contrast, the correlation between metabolic differences detected in brain and the other tissues was weaker ($p<10^{-8}$ for brain tissues and $p>10^{-4}$ for other tissues) ($p<10^{-8}$; *Figure 5—figure supplement 1c,d*). Thus, in the humanized mouse, the effects of the substitutions in ADSL are seen in the cerebral cortex and cerebellum in both young and adult animals.

## Purine biosynthesis in the humanized mice

In the 12-week-old mice, we detected six metabolites within the purine biosynthesis pathway. In the humanized mice, four of these metabolites had lower concentrations than in the wild-type mice in the cerebral cortex and all of them had lower concentrations in the cerebellum (binomial test for cortex and cerebellum, p=0.04) (*Figure 5b*). Further, concentrations of all six purine biosynthesis metabolites were significantly lower in the cerebellum of 12-week-old mice compared to metabolites assigned to other metabolic pathways (Wilcoxon test, p=0.04; *Figure 5c*).

The six metabolites detected in 12-week-old mice were also detected in the primate brains. Four of these had lower concentrations in the human brain, including AMP and GMP in prefrontal cortex, the visual cortex, and in cerebellum (binomial test, p=0.04). Furthermore, four of the six metabolites with lower concentrations in cerebellum in the humanized mice had lower concentrations in human cerebellum compared to other primates: AMP, GMP, inosine, and adenine (*Figures 2b* and *5b*). In the 1-week-old mice, we detected seven metabolites in the purine biosynthesis pathway. Five of these were present at lower concentrations in the cerebral cortex in the humanized mice and four of these at lower concentrations in the cerebellum when compared to their wild-type litter mates (*Figure 5b*).

Thus, several changes in concentrations of compounds in purine biosynthesis seen in the humanized mice recapitulate differences seen when the metabolomes of human brains are compared to the brains of chimpanzees and macaques.

## Activity and stability of humanized mouse ADSL

To investigate how the humanized form of the mouse ADSL enzyme may influence purine biosynthesis, we synthesized mouse wild-type (wt) ADSL and mouse A429V ADSL and inserted them in expression vectors that include N-terminal polyhistidine tags (pET-14b vector) (*Lee and Colman, 2007*). We analyzed the conversion of SAICAR to AICAR, and the conversion of SAMP to AMP, in the presence of excess substrate by measuring the rate of production of AICAR and AMP (*Figure 6a*) and found no differences in the kinetics of either reaction between wt and A429V ADSL (t-test, p>0.05).

We next investigated if the A429V substitution, and/or the R428Q substitution that was introduced into the humanized mice together with the A429V substitution, affect the conformational stability of the protein by constructing expression vectors that carry wild-type murine ADSL, ADSL with only the A429V substitution, ADSL with only an R428Q substitution, and ADSL with the A429V substitution and the adjacent R428Q substitution. We then compared the secondary structure stability of all four protein variants by measuring the circular dichroism spectra of the purified proteins at 222 nm while heating them from 55°C to 80°C at a rate of one degree per minute (*Figure 6b*). Mouse ADSL protein with alanine at position 429 was more stable (50% folded at 68.3°C,±0.2°C, n = 5) than mouse ADSL protein with valine at position 429 (50% folded at 67.9°C ± 0.2°C, n = 5). The effect on the protein stability of the A429V substitution was not influenced by the presence or absence of the adjacent R428Q substitution at position 428. In summary, the A429V substitution does not affect the kinetics of the murine ADSL enzyme. However, it decreases the secondary structure stability of the protein.

## Characterization of human and Neandertal ADSL

Previous work has shown that the modern human version of ADSL is less stable than the ancestral, Neandertal-like form in vitro (*Van Laer et al., 2018*). To ensure that the two forms of the ADSL protein which differ only at position 429 in the protein are both enzymatically active in living cells we show that they both are able to rescue Chinese hamster ovary cells that lack ADSL activity (*Adel* cells, *Vliet et al., 2011*; data not shown). We then expressed the modern human and Neandertal versions of ADSL in *E. coli* and isolated the two proteins by N-terminal His-tags (*Lee and Colman*,

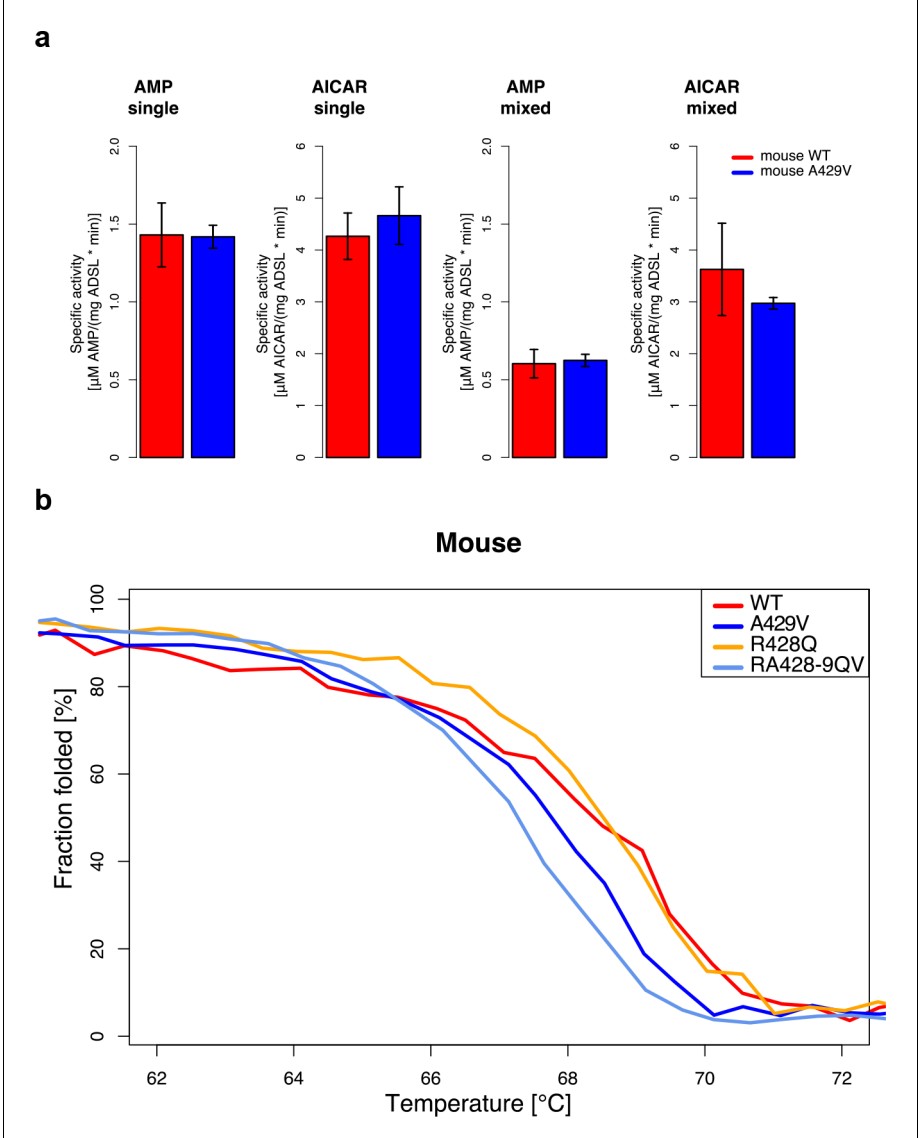

**Figure 6.** Characterization of mouse ADSL forms. (**a**) Enzyme kinetics tested in substrate excess for the products (AICAR and AMP, see text above the plots) with one substrate (single) or with both substrates (mixed) in the reaction mix. Specific activities of wild type and humanized A429V versions do not differ (t-test, p>0.05). Error bars represent standard error of the mean from three experiments using two batches of proteins. (**b**) Protein melting measured by CD at 222 nm for WT, A429V, R428Q, and RA428-9QV ADSL versions. Lines represent the averages over four experiments. The A429V (t-test, p=0.10) and RA428-9QV versions are less stable than the other proteins (t-test, p=0.03, at denaturation midpoint (dm). dm(WT)=68.3 +/- 0.2°C, dm(A429V)=67.9 +/- 0.2°C, dm(R428Q) =68.6 +/- 0.0°C, and dm(RA428-9QV)=67.3 +/- 0.3°C).

*2007*). In agreement with previous results (*Van Laer et al., 2018*), circular dichroism spectra of both protein versions exhibited predominantly alpha-helical characteristics (data not shown). To investigate if the enzymatic activities of the human and Neandertal versions of ADSL differ, we tested their ability to convert SAICAR to AICAR, and SAMP to AMP, as described above. No differences in specific activity between the two enzymes were detected using either substrate (SAMP, t-test, p=0.41 or SAICAR, t-test, p=0.81, *Figure 7a*).

To investigate if cooperativity between the two activities of ADSL may be affected by the A429V substitution, we incubated the two forms of the enzyme with equimolar amounts of SAMP and SAI-CAR and calculated relative values of ADSL protein activity when presented with a single substrate (*Figure 7a*) or an equimolar mixture of SAMP and SAICAR (*Figure 7b*). The conversion of SAMP to

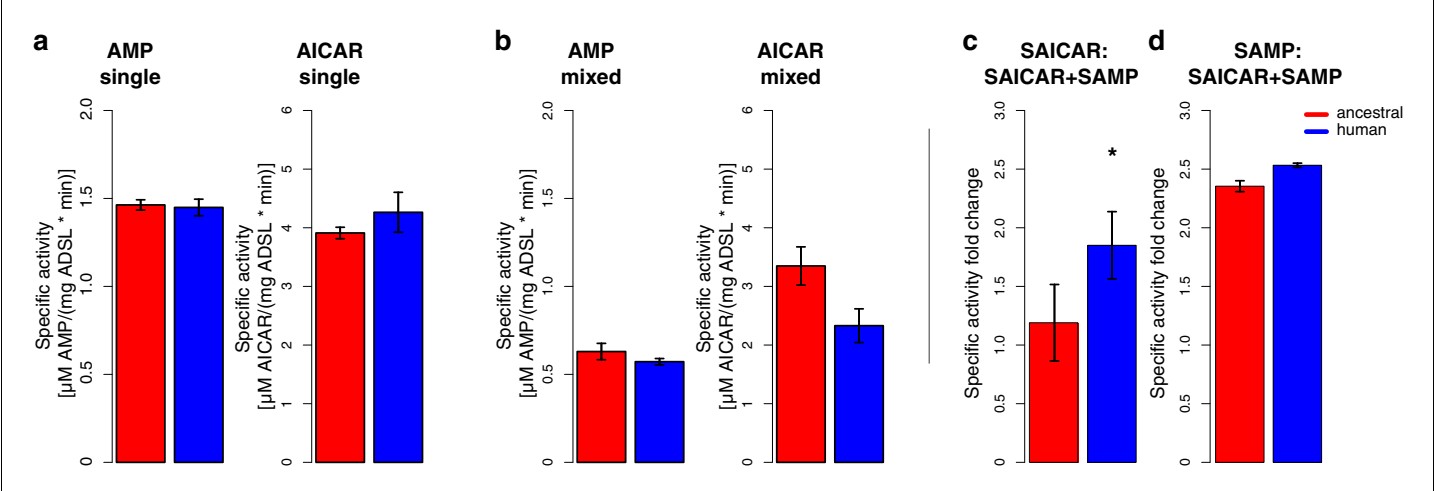

**Figure 7.** Specific activity and cooperativity of human and ancestral ADSL versions. Specific activity of SAMP to AMP and SAICAR to AICAR catalysis for the Neandertal-like and modern human forms of ADSL when incubated (**a**) with the required substrate separately ('single') (t-test, p>0.05); (**b**) and with both substrates (SAICAR and SAMP) together ('mixed') (t-test, p>0.05). (**c, d**) The ratios of single substrate to mixed substrate specific activities for the ancestral and human versions of the enzyme. The human version shows a higher reduction in specific activity for SAICAR to AICAR catalysis in the presence of both substrates than the ancestral version (t-test, p=0.01) (**c**) which is not seen for SAMP to AMP conversion (**d**) (t-test, p=0.49). Error bars represent standard errors among three experiments.

AMP is reduced approximately to the same extent (~2.4-fold, t-test, p=0.49) for the modern human and Neandertal-like forms of the enzyme when both substrates are present (*Figure 7d*). In contrast, the conversion of SAICAR to AICAR is reduced about 1.8-fold for the modern human form in the presence of both substrates (p=0.026) while the ancestral form of the enzyme shows no reduction (p=0.44; *Figure 7c*). This suggests that the modern human enzyme produces less AICAR than the ancestral enzyme under physiological conditions and may warrant further investigation.

## Stability of modern human and ancestral ADSL

To investigate the effects of the A429V substitution in the human protein, we compared the stability of the secondary structure of the protein as described above. The temperature at which 50% of the protein is inferred to be folded is 67.0°C (± SEM = 0.10°C) for the modern human form of the protein and 68.0°C (± SEM = 0.2°C) for the Neandertal-like form of the protein (*Figure 8a*). Thus, as previously shown (*Van Laer et al., 2018*), modern human ADSL is less stable than ancestral ADSL.

To investigate the effect of the difference in secondary structure stability on the stability of ADSL quaternary structures, we exposed the proteins to 0.75M, 0.90M, 1.0125M, 1.125M, and 1.5M guanidine hydrochloride (GdnHCl) overnight at room temperature and analyzed the proteins by native gel electrophoresis. Bands corresponding to the predicted size of the ADSL tetramer (~223 kDa) and monomer (~56 kDa) were observed and the relative proportion of tetrameric protein was estimated. *Figure 8b* shows that at 1.0125 M GdnHCl about 50% of the modern human version of ADSL was present as tetramers whereas for the ancestral form this was the case at 1.125 M GdnHCl. Thus, both the secondary and quaternary structures of the ancestral form of ADSL are more stable than for the modern human form.

## Ancestralized ADSL in human cells

To investigate how the A429V substitution affects the metabolome of living human cells, we used CRISPR-*Cas9* to introduce the nucleotide substitution in the *ADSL* gene that reverts the valine at position 429 to the ancestral alanine residue in human 409B2 cells (Riken BioResource Center). We isolated three independent cell lines and we verified that the intended nucleotide substitution had occurred in each of them by sequencing a segment of the *ADSL* gene and excluded deletions of the target site by quantitative PCR (data not shown). We also isolated six independent lines that had

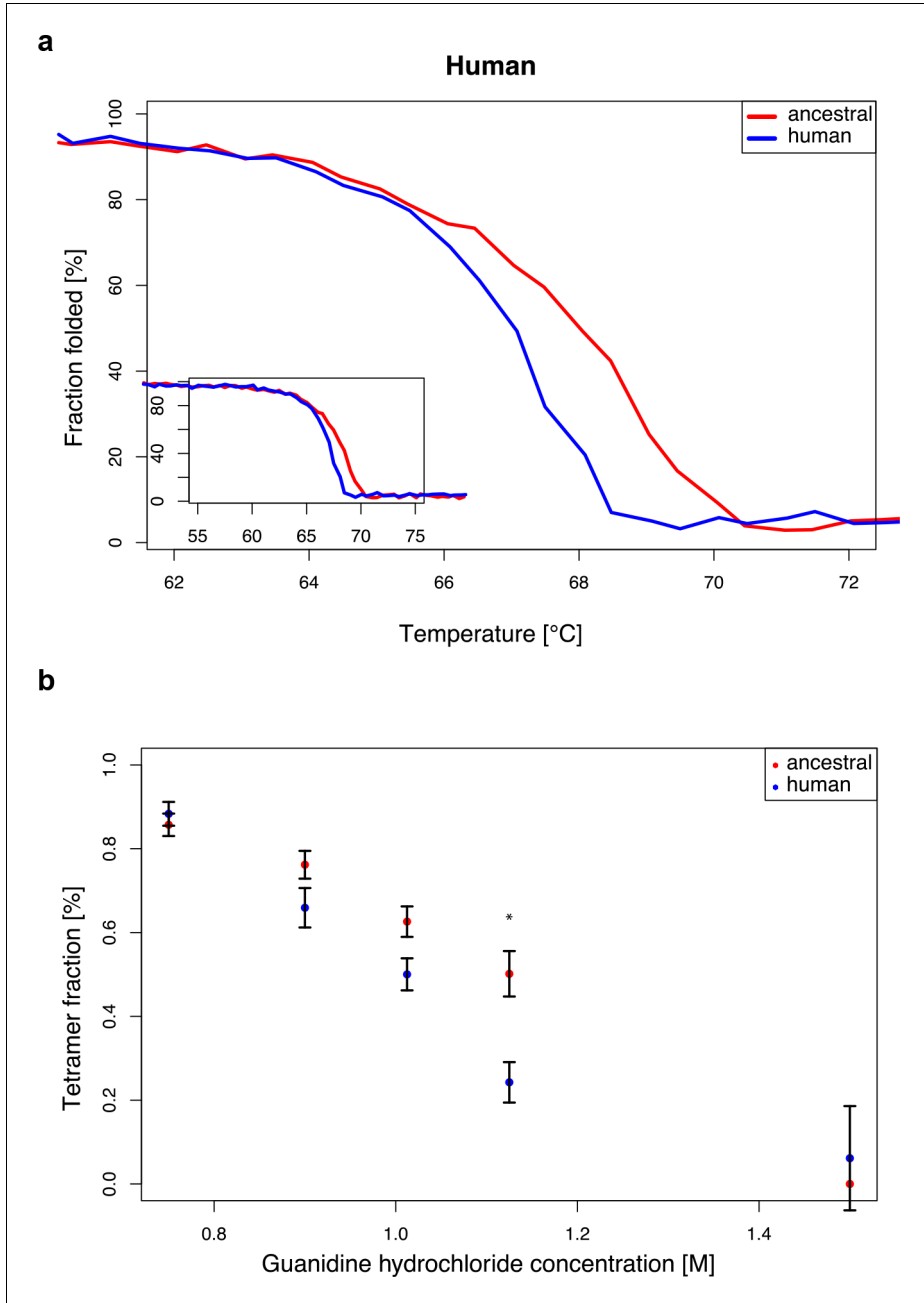

**Figure 8.** Stability of modern human and Neandertal-like ADSL. (**a**) Thermal denaturation measured by CD at 222 nm for 55–75°C (plot insert) and enlarged for 62–72°C (bigger panel) showing lower stability of the modern human ADSL at the denaturation midpoint (t-test, p=0.001. dm(ancestral)=68.0 +/- 0.2°C, dm(human)=67.0 +/- 0.1°C). Data represents the average of five experiments. (**b**) Chemical denaturation measured by native gel electrophoresis (p=0.024 for 1.125M Gdn-Hcl concentration, t-test with Bonferroni correction). Plot represents average amount of tetrameric protein over five experiments. Error bars represent standard errors.

been subjected to the CRISPR-*Cas9* procedure but did not exhibit any mutation in the sequenced DNA segment. We expanded 10 separate cultures of the three edited cell lines and 19 cultures of three control lines (*Figure 9—figure supplement 1*). Each of these cultures were analyzed by LC-MS in positive and negative modes.

A total of 8849 metabolite peaks were detected in positive and negative mode (*Supplementary file 10*). Among them, 1989 were computationally annotated using KEGG.

Strikingly, out of 10 detected metabolites downstream of ADSL, all but one (adenine) have lower concentrations in the wild type cells (binom. test, p=0.01; Wilcoxon test, p=0.06, *Figure 9a*). These metabolites are present in lower amounts in the wild-type human cells than the cell carrying the V429A substitution in ADSL (Wilcoxon test, p=0.06; *Figure 9b*). Thus, in human cells, the ancestral form of ADSL supports a higher level of purine biosynthesis than the modern human form.

## Comparison to chimpanzee and macaque cells

To compare these results to purine biosynthesis in chimpanzees and macaques, we similarly analyzed nine cultures of three different chimpanzee pluripotent cell lines and three cultures of a macaque pluripotent cell line (*Figure 9—figure supplement 1*). Out of 1883 metabolite peaks that differ significantly (Wilcoxon test, BH-adjusted p<0.05) between the human and chimpanzee wild-type cells, four correspond to purine biosynthesis metabolites (guanosine, inosine, hypoxanthine, xanthine) and are present at lower levels in human cells compared to chimpanzee cells. The same 10 metabolites as in the comparison to the ancestralized human cells were detected in the chimpanzee cells. Seven of them are present in lower amounts in the human cells (*Figure 9a*) and as a group they are present in lower concentrations in the human than in the chimpanzee cells when compared to metabolites in other pathways (Wilcox test, p=0.01, *Figure 9b*).

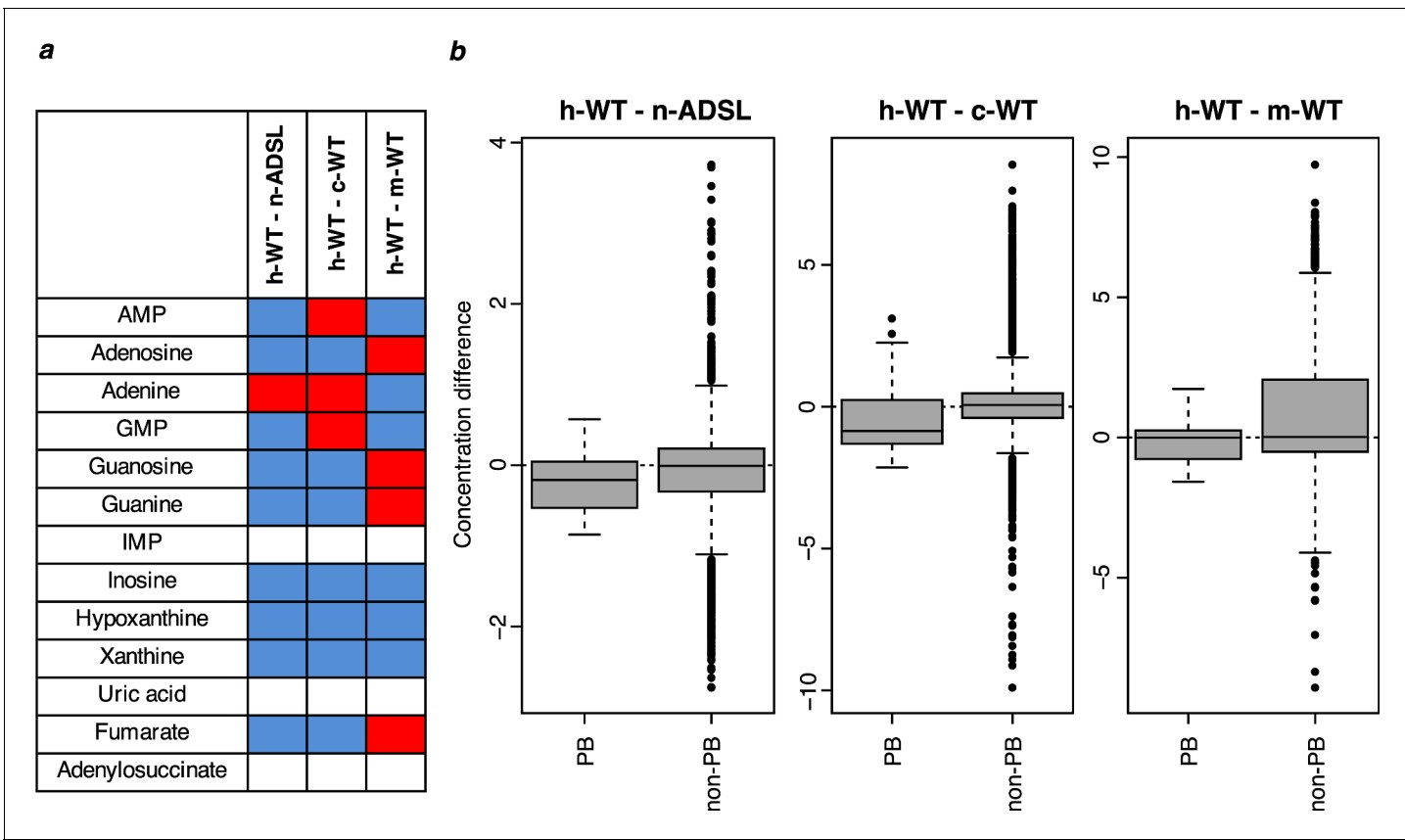

**Figure 9.** Purine biosynthesis in human cells carrying the ADSL V429A substitution as well as in human, chimpanzee, and macaque wild type cells. (**a**) The direction of differences in metabolite levels in pairwise comparisons between wild type human cells (h-WT, 19 cultures of three cell lines) and human cells carrying the ADSL V429A substitution (n-ADSL, 10 cultures of three cell lines), chimpanzee cells (c-WT, nine cultures from three cell lines) and macaque cells (m-WT, three cultures from one cell line). Red and blue colors mark higher and lower metabolite levels in human wild-type cells, respectively, white indicates non-detected metabolites. (**b**) Distributions of differences in metabolite levels between h-WT cells and n-ADSL, c-WT, and m-WT cells, respectively, for purines biosynthesis (PB) metabolites (listed in panel a) and other metabolites (non-PB) Wilcoxon test, p=0.06 (h-ADSL), p=0.01 (c-WT), and p=0.04 (m-WT).

The online version of this article includes the following figure supplement(s) for figure 9:

**Figure supplement 1.** Principle component analysis of cell line metabolite composition.

Six of the 10 purine biosynthesis metabolites are present at significantly lower concentrations in humans than in macaques when analyzed individually (*Figure 9a*) and as a group they are present at lower concentrations than metabolites in other pathways (Wilcoxon test, p=0.04, *Figure 9b*). Thus, reversal of the A429V substitution in ADSL in human cells results in an increase in purine biosynthesis similar in magnitude to what is observed when chimpanzee cells are compared to human cells.

## Discussion

The ancestors of modern humans diverged from their closest evolutionary relatives, Neandertals and Denisovans, on the order of 600,000 years ago (*Prüfer et al., 2014*). About 10 times further back the ancestors of hominins diverged from the ancestors of chimpanzees and bonobos (*Langergraber et al., 2012*). Whereas some phenotypic differences between present-day humans and apes have been shown to be due to genetic features unique to humans, especially gene duplications (e.g. *Charrier et al., 2012*; *Nuttle et al., 2016*; *Florio et al., 2018*) and changes involving gene expression (reviewed by *Doan et al., 2018*), no organismal traits that distinguish present-day modern humans from Neandertals and Denisovans have been associated with specific nucleotide changes. Whereas most phenotypic features that distinguish present-day humans from the apes and from their extinct hominin relatives are likely to be genetically complex, metabolic differences may sometimes have a comparatively simple genetic background because genomic changes that affect genes encoding enzymes may affect flux in metabolic pathways.

To find metabolic differences that set humans apart from their closest evolutionary relatives we investigated the metabolomes of the brain, muscle, and kidney in humans, apes, and monkeys. Whereas we find no such differences among the metabolites analyzed in muscle and kidney, in the brain, steady state abundance of many compounds involved in amino acid metabolism are present in higher or lower levels in humans versus other primates and metabolites involved in oxidative phosphorylation are present in lower amounts. In the future, it may be of interest to investigate more samples and the consequences of these human-specific metabolic features. Metabolites in purine biosynthesis stand out in that they are present in lower concentrations in humans than in the other primates analyzed in all tissues analyzed, although most drastically in brain.

Humans and apes diverged so long ago that almost every gene carries changes that may potentially alter its function by affecting its regulation or the structure of the encoded gene product. In contrast, modern humans and Neandertals and Denisovans diverged so recently that for about 90% of the genome, the two archaic human groups fall within the variation of present-day humans (*Green et al., 2010*). Furthermore, when modern and archaic humans met about 50,000 years ago, they interbred. This resulted in that Neandertal DNA fragments that together make up about half of the Neandertal genome exist in present-day humans (*Sankararaman et al., 2014*). The number of proteins that carry amino acid substitutions in all or almost all humans that differ from Neandertals and apes is therefore only about one hundred (*Pääbo, 2014*). It is unclear if any of these substitutions have any functional consequences.

The alanine to valine substitution at position 429 in ADSL is one of this small number of substitutions. Position 429 is conserved as alanine in most tetrapods suggesting that it may be of importance. Position 429 is also located only three positions away from position 426, where an arginine to histidine substitution causes the most common form of adenylosuccinase deficiency in present-day humans (*Edery et al., 2003*; *Kmoch, 2000*; *Maaswinkel-Mooij et al., 1997*; *Marie et al., 1999*; *Race et al., 2000*). Further evidence suggesting that a change in *ADSL* may have been of importance in the evolution of modern humans comes from a screen for genomic regions that have experienced selective sweeps in humans after their split from Neandertals but before the separation of Africans and Eurasians (*Racimo, 2016*). In that study, a genomic region centered around *ADSL* is among the top 20 candidate regions, although it also contains other genes. Furthermore, previous work has shown that the A429V substitution reduces the thermal stability of the ADSL protein in vitro (*Van Laer et al., 2018*). We therefore decided to analyze if it might be involved in the reduced purine biosynthesis seen in present-day humans by investigating the function of the ancestral, Neandertal-like and the derived, modern human-like forms of ADSL in vitro and in vivo.

We confirm the previous finding (*Van Laer et al., 2018*) that the A429V substitution does not affect the kinetic properties of the ADSL enzyme but decreases its thermal stability and show that the substitution also decreases the stability of the tetrameric complex of the enzyme. When

introduced in the mouse ADSL protein in conjunction with a primate-specific substitution at the adjacent position 428, it reduces the enzymatic activity detected in nine tissues analyzed, most drastically in the brain, and results in a reduction in purine biosynthesis, thus recapitulating differences seen between humans and chimpanzees and macaques.

To investigate how the A429V substitution may affect the metabolism of human cells, we used CRISPR-*Cas9* to introduce the ancestral, Neandertal-like substitution into human cells. The concentrations of nine out of 10 detected metabolites downstream of ADSL in purine biosynthesis are increased in human cells carrying the ancestral substitution. In chimpanzee and macaque cells, most of the same 10 metabolites are similarly increased resulting in that the pathway as a whole is increased relative to other pathways. Notably, the expression of ADSL messenger RNA does not differ between human and chimpanzee cells, nor between wild type and ancestralized cells (not shown). Thus, the A429V substitution is responsible for much or all of the difference in purine biosynthesis observed when human tissues are compared to ape and monkey tissues, indicating that this change in metabolism occurred in humans after their separation from the ancestor shared with Neandertals and Denisovans.

It is interesting that although ADSL is expressed and functions in all tissues, the down-regulation of purine biosynthesis in humans relative to apes, and in humanized mice relative to wild-type mice, is most pronounced in the brain. It is also interesting that mutations in humans that affect enzymes involved in purine metabolism have more pathological consequences in the nervous system than in other organs (*Fumagalli et al., 2017*; *Micheli et al., 2011*). Although no overt morphological or other brain-related phenotypes are observed in the mice humanized at pos. 428 and 429 (data not shown), it is thus possible that the A429V substitution in ADSL has contributed to human-specific changes in brain development and function. Future work will have to address this and other possibilities.

## Materials and methods

### Tissue samples

Human samples were obtained from the Netherlands Brain Bank, Amsterdam, Netherlands. Written consent for the use of human tissues for research was obtained from all donors or their next of kin. All subjects were defined as healthy controls by forensic pathologists at the brain bank. All subjects suffered sudden death with no prolonged agonal state. Chimpanzee samples were from the tissue collection of the Max Planck Institute for Anthropology. Rhesus macaque samples were obtained from the Suzhou Experimental Animal Center, China. All chimpanzees and rhesus macaques used in this study suffered sudden deaths for reasons other than their participation in this study and without any relation to the tissue used. Prefrontal cortex (PFC) dissections were made from the frontal part of the superior frontal gyrus. For all samples, we took special care to dissect gray matter only.

### Primate cell lines

Three chimpanzee iPSC lines used were: Sandra-A, generated from T lymphocytes *Mora-Bermudez et al., 2016*; JoC generated from primary lymphocytes *Kanton et al., 2019*; and PR818-5 generated from fibroblast (*Marchetto et al., 2013*), a gift from Fred H. Gage, the Salk Institute for Biological Studies (La Jolla, CA). The macaque embryonic stem cell line MN1 (*Otani et al., 2016*) was a gift from R. Livesey, Gurdon Institute (University of Cambridge, UK). All lines were cultured in six-well cell culture plates coated with Matrigel (Corning, 354277) and maintained in mTeSR1 medium (Stem Cell Technologies, 05851) supplemented with penicillin/streptomycin and MycoZap (Lonza) at 37°C in a humidified 5% $CO_2$ incubator. The medium was refreshed daily. All cell lines were regularly checked for mucoplasma.

### Genome editing in mice

Male pronuclei of C57BL/6NCrl fertilized oocytes were injected with a mix consisting of Cas9 or Cas9-D10A Nickase mRNA (50 ng/μl), gRNA in form of RNA (20 ng/μl) and single strand oligonucleotide (50 ng/μl). The sequence of the 99mer oligonucleotide is 5'-GTGGTCAAGCAGGAAGGAGG TGACAATGACCTTATAGAGCGCATC**CAGGTT**GACGCCTACTTCAGCCCCATCCACTCACAGC TGGAGCACTTGCTGGAC-3'. The nucleotide substitutions for changing the amino acids arginine

(CGG) and alanine (GCA) to glutamine (CAG) and valine (GTT) that occur in the human sequence are depicted in bold. The humanized sequence introduces two new restriction sites for *BstNI* and *HincII* that facilitate genotyping of the founders. The sequence of the two guide RNAs are: Adsl-gRNA10: 5′-GGCTGAAGTAGGCATCTGCC-3′ for Cas9 and Adsl-gRNA15: 5′-CTCACAGCTGGAGCACTTGC-3′ used together with gRNA10 for the Cas9-D10A injections. After injections, the surviving embryos were transferred into Crl:CD1(ICR) pseudopregnant recipient female mice (20–25 embryos per recipient). Tail DNA of the founders was PCR amplified with Adsl-F: 5′-CGATGTCTGACTGTAAGGTC TAC-3′ and Adsl-R: 5′-CCAGAGGCTCAGGCTCTGCATCA-3′ and digested with *HincII*. In total 6 from 105 born mice were positive for the humanized sequence.

## Genome editing in human cells

409-B2 human induced pluripotent stem cells (hiPSCs) with a doxycycline inducible Cas9 nickase (D10A mutation) and enhanced homology-directed repair efficiency (DNA-PKcs K3753R mutation) were incubated with mTesR1 medium (StemCell Technologies, 05851) containing 2 µg/ml doxycycline (Clontech, 631311) two days prior to lipofection of a gRNA (duplex of chemically synthesized crRNA and tracrRNA, alt-CRISPR IDT) pair and single-stranded DNA donor as described (*Riesenberg et al., 2019*). Lipofection by RNAiMAX (Invitrogen, 13778075) was done using a final concentration of 7.5 nM of each gRNA (ADSL_target1: CAACCTGGATACGCTCTATG, ADSL_target2: CAGTCCCATTCACTCCCAGT) and 10 nM of single-stranded DNA donor (ADSL_V429A: CAG-CAGGCAGCTTCTGTGGTTAAGCAGGAAGGGGGTGACAATGACCTCATTGAGCGTATCCAGGC TGATGCCTACTTCAGTCCCATTCACTCTCAGTTGGATCATTTACTGGATCCTTCTTCTTTCACTGG TCGTGCCTCCCAG; ancestral mutation = bold, silent mutations to prevent recutting: italic). The lipofection mix was exchanged to regular mTesR1 medium after 24 hr and cells were propagated. After dissociation using Accutase (SIGMA, A6964) cells were plated in a single-cell dilution that gave rise to single-cell derived colonies. DNA isolation, Illumina library preparation and sequencing analysis to conform editing success was done as described (*Riesenberg et al., 2019*). Primers for DNA target amplification and quantitative PCR were ADSL_forward: AAAGTGTTCAAACGCCCTGT and ADSL_-reverse: AGAGAGTACCCCAGGTTCTGC.

## Enzyme assays

Tissue samples were homogenized in 300 mM sucrose, 10 mM Tris pH 7.4, 10 mM EDTA using a PTFE tissue grinder and mixer motor. Samples were centrifuged at 4°C for 30 min in a microfuge at maximum speed. Supernatant was collected and stored at −70°C. Protein concentration was determined using a bicinchoninic acid assay kit (Sigma) and a BioTek PowerWave XS2 plate reader, measuring absorbance at $\lambda$ 562 nm. Absorbance values were fitted against a BSA standard curve (0, 12.5, 25, 50, 75, 100, 150, 200 µg/ml). For the ADSL assays, 50 ng protein was added into a 60 µM adenylosuccinate (AMPS)/40 mM Tris (pH 7.4)/10% glycerol solution, and conversion of adenylosuccinate to AMP was monitored by UV spectrophotometry, measuring absorption at $\lambda$ 282 nm every minute for 20 min (Greiner UV-Star 96 well plates and BioTek PowerWave XS2 plate reader). ADSL activity was calculated using the extinction coefficient for the reaction (10,000 M-1 cm-1) in Excel. Wilcoxon tests and box plots were performed using R.

## CE-MS measurements of metabolomes

For each sample, metabolites were extracted from the frozen tissue powder by 1 ml methanol containing 20 µM each of L-Methionine sulfone, 2-morpholinoethanesulfonic acid, monohydrate, and sodium d-camphor-10-sulfonic acid. 500 µl of the lysate was transferred to an Eppendorf tube containing 500 µl chloroform and 200 µl of Milli-Q water. After 30 s of vortexing and 15 min of centrifugation at 4°C, 300 µl of the aqueous phase was concentrated and dried to complete dryness in an ultrafiltration tube (Millipore) followed by speed vacuum for 3 hr at 35°C. The dried samples were mixed with 100 µl of Milli-Q water containing 100 µM each of 3-aminopyrrolidine and trimesate, and filtered with an ultrafiltration tube (Millipore) at 9100 × g for 2 hr at 4°C immediately before 7 µl was used in a CE-TOF-MS (Agilent Technologies) to detect cationic metabolites and anionic metabolites. The instrumentation and measurement conditions used for CE-TOF-MS were according to *Sugimoto et al., 2012*.

The in-house software MasterHands was used for peak detection, time alignment, and peak area integration. The intensities of each metabolite in the samples were calculated based on the comparison of peak area normalized by internal standards added to the sample as well as external standards. Metabolites detected in less than half of the samples were excluded from the following analysis.

## GC-MS measurements of metabolomes

Metabolites were extracted from the frozen tissue powder by a methanol:water:chloroform (2.5:1:1 (v/v/v)) extraction. In brief, 100 mg of frozen powdered tissue material was resuspended in 1 mL extraction solution containing 0.1 µg mL-1 of U-13C6-sorbitol. The samples were incubated for 10 min at 4°C on an orbital shaker. This step was followed by ultrasonication in a bath-type sonicator for 10 min at room temperature. Finally, the insoluble tissue material was pelleted by a centrifugation step (5 min; 14,000 g), and the supernatant transferred to a fresh 2 mL Eppendorf tube. To separate the organic from the aqueous phase, 300 µL $H_2O$ and 300 µL chloroform were added to the supernatant, vortexed, and centrifuged (2 min; 14,000 g). Subsequently, 200 µL of the upper, aqueous phase was collected and concentrated to complete dryness in a speed vacuum at room temperature. Extract derivatization and GC-MS measurements were performed according to *Lisec et al., 2006*.

MS peaks were aligned across samples and annotated to known or putatively known compounds, based on similarity searches against spectral libraries of several thousand measured reference compounds (*Giavalisco et al., 2011*; *Cuadros-Inostroza et al., 2009*). Unannotated MS peaks were excluded, and each tissue was further analyzed separately. The obtained metabolite intensity values (apex height of the quantitative compound identifier mass) were normalized within each sample to the abundance of an internal standard (13C sorbitol), and log10 transformed. To avoid negative values, metabolite/standard ratios were scaled up by factor 3000 prior to log10 transformation. Next, we filtered out metabolites containing more or equal to 50% missing values. To normalize distributions of values between samples, we additionally divided intensity values for the remaining metabolites to the upper quartiles.

## LC-MS sample preparation

Adhered cells were grown mTesR1 medium (StemCell Technologies, 05851) in six-well plates. After counting, medium solution was gently discarded and cells were quickly washed twice with 1 ml of 1x PBS. Cells were harvested in PBS, placed into 2 ml Eppendorf (safe lock) tubes, suspended by slight pipetting, pelleted by centrifugation at 4°C and snap-frozen in liquid nitrogen. Pellets were stored at −80°C. The several blank samples were added to the end of each batch (48 samples). They were 2 ml Eppendorf (safe lock) tubes without any sample material but subjected to all extraction procedures. Cells were quickly re-suspended in 75 µl of water on ice and metabolites were extracted by the after addition of 1 ml of −20°C cold methanol:methyl-tert-butyl-ether (1:3 (v/v)) extraction buffer, vortexed, and sonicated for 10 min in an ice-cooled sonication bath followed by incubation for a 30 min at 4°C on an orbital shaker as described (*Giavalisco et al., 2011*). A total of 700 µl of a H2O: methanol mix (3:1 (v/v)), containing 5.0 µg of GMP-15N5 and Methionine-methyl-13C.d3 was added and the mixture was incubated for a 10 min at 4°C on an orbital shaker. To separate the organic and the aqueous phases and precipitate proteins the mixture was spun down for 10 min at 15,000 x g at 4°C. Subsequently, 300 µl of the lower aqueous phase containing hydrophilic compounds was transferred to a 1.5 ml Eppendorf tube and concentrated to complete dryness in a speed vacuum centrifuge at room temperature. Dry metabolic extracts were stored at −80°C prior to mass spectrometry analysis.

In the LC-MS datasets, in addition to the individual cells samples, we measured mixtures of samples (pooled extracted samples, TQC) after every 10th sample, providing information on system performance (sensitivity and retention time consistency), sample reproducibility, and compound stability over the time of the MS-based analysis.

## LC-MS metabolite abundance measurements

The dried extracts were re-suspended in 100 µl of ice-cold 20% aqueous solution of acetonitrile prior to mass spectrometry analysis. After brief rigorous vortexing, the samples were incubated for 30 min at 4°C on an orbital shaker followed by a 10 min ultra-sonication in an ice-cooled sonication bath and

centrifugation for 10 min at 15,000 x g at 4°C. For mass spectrometry analysis, 40 µl of supernatant was transferred to a 350 µl autosampler glass vials (Glastechnik Grafenroda, Germany). Chromatographic separation of metabolites prior to mass spectrometry was performed using Acquity I-Class UPLC system (Waters, UK). Metabolites were separated on a normal phase unbounded silica column RX-SIL (100 mm x 2.1 mm, 1.8 µm, Agilent, US) coupled to a guard precolumn with the same phase parameters. The mobile phases used for the chromatographic separation were water containing 10 mM ammonium acetate, 0.2 mM ammonium hydroxide in water:acetonitrile (95:5 (v:v)) mixture (buffer A) (pH value 8.0) and 100% acetonitrile (buffer B). The gradient separation was: 0 min 0% buffer A, 0.01–15 min linear gradient from 0% to 100% buffer A, 15–18 min 100% buffer A, 18–19 min linear gradient from 100% buffer A to 0% buffer A, and 19–32 min 0% buffer A. After 1 min washing with 100% buffer A, the column was re-equilibrated with 100% buffer B. The flow rate was set to 500 µl/min. The column temperature was maintained at 32°C. The mass spectra were acquired in positive and negative mode using a heated electrospray ionization source in combination with Q Exactive Hybrid Quadrupole-Orbitrap mass spectrometer (Thermo Scientific. Germany). Negative ion mode samples were run after the positive ion mode cohort with 6 µl injection of non-diluted samples. MS settings in positive acquisition mode: spray voltage was set to 4.5 kV in positive mode and to 3 kV in negative acquisition mode. S-lens RF level at 70. and heated capillary at 250°C; aux gas heater temperature was set at 350°C; sheath gas flow rate was set to 45 arbitrary units; aux gas flow rate was set to 10 arbitrary units; sweep gas flow rate was set to four arbitrary units. Full scan resolutions were set to 70,000 at m/z 200. Full scan target was $10^{-6}$ with a maximum fill time of 50 ms. The spectra were recorded using full scan mode, covering a mass range from 100 to 1500 m/z.

For quality control (TQC), a pooled sample of all metabolic extracts was prepared and injected four times before initiating the runs in order to condition the column, at least four times after each sub-cohort, and after the completion of the runs. In addition, the TQC sample was injected every 48 sample injections to assess instrument stability and reproducibility.

Metabolite peaks were extracted and aligned across samples using Progenesis QI software (Version 3.0, Nonlinear Dynamics, http://www.nonlinear.com) according to the vendor description. We further excluded metabolite peaks potentially confounded by their processing order during mass spectrometry measurements (run order) using a support vector regression model with a Gaussian kernel. The intensities of metabolites retained after the above-mentioned filtration procedures were log2 transformed and normalized using upper-quartile normalization procedure.

## Statistical analysis of metabolomes

We performed principal component analysis (PCA) for each tissue separately or for cell line data in order to identify outlier samples. For the PCA analysis, the 'prcomp' function in the R package 'stats' was used. To identify metabolites with significant intensity changes, we implemented the Student's t-test. A permutation procedure was used to assess the false positive rate. Briefly, we shuffled sample labels, applied t-test to two random groups of samples, and repeated this procedure 1000 times. Then, a permutation p-value was used to estimate the ratio of permutations resulting in an equal or greater number of metabolites with significant intensity differences at a nominal significance cutoff of 0.05. To determine the probability of increase or decrease of metabolite intensities, we performed a binomial test. Log10 fold change of metabolite intensities was calculated as the average logarithm of intensity differences between two sample groups. We used Wilcoxon test to compare the distributions of intensity differences between species or mutant and wild type individuals for detected metabolites assigned to purine biosynthesis pathways and those assigned to the other pathways.

## Protein purification and dual substrate specific activity measurement

ADSL protein versions were purified as described previously (*Ray et al., 2012*). The specific activity in presence of single and double substrates shown in *Figure 6* was measured using a customized HPLC system as described previously (*Ray et al., 2013*).

## Native gel electrophoresis

Aliquots of ADSL protein were thawed and equilibrated for 2–4 hr at room temperature. Subsequently, they were mixed with guanidine hydrochloride to final concentrations of 1.5M, 1.125M,

1.0125M, 0.9M, and 0.75M. Samples were incubated overnight at room temperature and analyzed using native gel electrophoresis (Invitrogen, BN2111B × 10, NativePage 4–16%) according to manufacturer's manual. ImageJ was used for densitometry analysis of monomeric (M) and tetrameric (T) band of gel pictures where person assigning the peaks was blinded for the sample identity. Tetramer fraction was subsequently calculated using the following formula: T/(M + T). Differences between human and Neandertal-like versions of ADSL were calculated for each concentration of guanidine hydrochloride over five repetitions. Bonferroni correction was used to correct for number of comparisons (five different guanidine hydrochloride concentrations).

### Circular dichroism measurements

ADSL versions were diluted in storage buffer as described to concentration of 0.2 mg/ml. Samples were thawed and equilibrated at room temperature for a minimum of 30 min prior to the scan. Each sample was transferred to a 1 mm cuvette for analysis. To analyse secondary structure, each sample was first scanned for wavelengths 200–250 nm and subsequently monitored at 222 nm from 50°C to 80°C using a Jasco J-815 Spectropolarimeter. Each ADSL protein version was analyzed using a minimum of two separate protein batches and a minimum of two scans to avoid batch effects. Denaturation midpoint was designated as the temperature value closest to the 50% folded protein mark for each run.

## Acknowledgements

We are grateful to the NOMIS Foundation and the Max Planck Society for funding to SP; to the Bonfils-Stanton Foundation, the Butler Family Fund of the Denver Foundation, the Sam and Freda Davis Charitable Trust, and the Theodore T Puck Endowment of the University of Denver for funding to DP; to Stephane Peyregne and Karin Mörl for helpful input, to Rowina Voigtländer, Wulf Hevers and the animal facility staff for mouse breeding, and to Alexander Cagan and Wulf Hevers for tissue dissections. Animal breeding and experiments were done under the permission AZ: 24–9162.11/12/12 (T 10/14) from the Landesdirektion Sachsen.

## Additional information

### Funding

| Funder | Author |
| --- | --- |
| NOMIS Stiftung | Svante Pääbo |
| Max-Planck-Gesellschaft | Svante Pääbo |
| Bonfils-Stanton Foundation | David Patterson |
| Denver Foundation | David Patterson |
| University of Denver | David Patterson |

The funders had no role in study design, data collection and interpretation, or the decision to submit the work for publication.

### Author contributions

Vita Stepanova, Guido N Vacano, Investigation, Methodology, Writing - original draft; Kaja Ewa Moczulska, Formal analysis, Investigation, Writing - original draft; Ilia Kurochkin, Stephan Riesenberg, Dominik Macak, Tomislav Maricic, Linda Dombrowski, Maria Schörnig, Ekaterina Khrameeva, Anna Vanushkina, Elena Stekolshchikova, Alina Egorova, Anna Tkachev, Randall Mazzarino, Nathan Duval, Dmitri Zubkov, Terry G Wilkinson, Investigation; Xiangchun Ju, Investigation, Writing - original draft; Konstantinos Anastassiadis, Oliver Baker, Ronald Naumann, Patrick Giavalisco, Investigation, Methodology; David Patterson, Conceptualization, Investigation, Project administration; Philipp Khaitovich, Conceptualization, Supervision, Investigation, Methodology, Project administration; Svante Pääbo, Conceptualization, Funding acquisition, Project administration, Writing - review and editing

## Author ORCIDs

Kaja Ewa Moczulska https://orcid.org/0000-0001-6051-1265
Guido N Vacano https://orcid.org/0000-0001-5979-9310
Ilia Kurochkin http://orcid.org/0000-0003-3100-0903
Maria Schörnig http://orcid.org/0000-0001-5334-5342
Konstantinos Anastassiadis http://orcid.org/0000-0002-9814-0559
Patrick Giavalisco http://orcid.org/0000-0002-4636-1827
Philipp Khaitovich https://orcid.org/0000-0002-4305-0054
Svante Pääbo https://orcid.org/0000-0002-4670-6311

## Ethics

Human subjects: Human postmortem samples were obtained from the NICHD Brain and Tissue Bank for Developmental Disorders at the University of Maryland, USA, the Maryland Brain Collection Center, Maryland, USA, and the Harvard Brain Tissue Resource Center. Informed consent for the use of human tissues for research was obtained by these institutions in writing from all donors or their next of kin.

Animal experimentation: Mouse breeding and experiments were done under the permission AZ: 24-9162.11/12/12 (T 10/14) from the Landesdirektion Sachsen. This study was reviewed and approved by the Institutional Animal Care and Use Ethics Committee at the Shanghai Institute for Biological Sciences, CAS. All non-human primates used in this study suffered sudden deaths for reasons other than their participation in this study and without any relation to the tissue used.

## Decision letter and Author response

Decision letter https://doi.org/10.7554/eLife.58741.sa1
Author response https://doi.org/10.7554/eLife.58741.sa2

# Additional files

## Supplementary files

• Supplementary file 1. The MS signal intensities of metabolites detected by CE-MS in five tissues of four humans, four chimpanzees, and four rhesus macaques, reflecting their relative concentrations.

• Supplementary file 2. Metabolites showing significantly higher intensities in the human cerebellum compared to chimpanzees and rhesus macaques (t-test, adjusted $p<0.05$). The table includes adjusted p-values and fold-change values for human-chimpanzee and human-macaque comparisons.

• Supplementary file 3. Pathway enrichment in metabolites showing higher levels in human tissues compared to chimpanzees and rhesus macaques. The table includes pathway names, adjusted and nominal p-values, number of differential metabolites, and number of associated genes. Top pathways (corrected enrichment p-value<0.05) are shaded in green. Tables include multiple worksheets – one for each tissue.

• Supplementary file 4. Metabolites showing significantly lower intensities in the human prefrontal cortex compared to chimpanzees and rhesus macaques (t-test, adjusted $p<0.05$). The table includes adjusted p-values and fold-change values for human-chimpanzee and human-macaque comparisons.

• Supplementary file 5. Pathway enrichment in metabolites showing lower levels in human tissues compared to chimpanzees and rhesus macaques. The table includes names, adjusted and nominal p-values, number of differential metabolites, and number of associated genes. Top pathways (corrected enrichment p-value<0.05) are shaded in green. Tables include multiple worksheets – one for each tissue.

• Supplementary file 6. Mouse sample information. Table includes sample information for two age groups of wild type and humanized mice: one week and 12 weeks of age. Eight tissues were sampled for both age groups. Testes were additionally sampled for 12-weeks-old mice. Outlier samples are shaded in red. Table includes multiple worksheets – one for each age group.

• Supplementary file 7. Normalized intensities of metabolites detected using GC-MS in eight tissues of 1-week-old humanized and wild-type mice. The table includes multiple worksheets – one for each tissue.

• Supplementary file 8. Normalized intensities of metabolites detected using GC-MS in eight tissues of 12-weeks-old humanized and wild-type mice. The table includes multiple worksheets – one for each tissue.

• Supplementary file 9. Metabolites showing significant intensity differences between humanized and wild-type mice. Table lists 45 metabolites showing significant intensity differences in the pre-frontal cortex (PFC) at one week of age and 36 showing significant intensity differences in the in cerebellum (CB) at 12 weeks of age. Table includes multiple worksheets – one for each tissue.

• Supplementary file 10. Normalized intensities of metabolites detected using LC-MS in cell lines. The table includes multiple worksheets: cell line metadata, intensities of all detected metabolite peaks, peak annotation according to KEGG database.

• Transparent reporting form

## Data availability
All data generated are included in the paper as Supplementary files 1-10.

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
