## [Decision Letter]

**Acceptance summary:**

This is an exciting study that functionally investigates how one trait, metabolism, may have evolved in modern humans. The authors show how one human-specific alteration affects purine biosynthesis through a series of comparisons among primates, the generation of humanized mice, and ancestral mutations in human cells. In particular, the metabolic changes are robust in brain tissue and future studies will evaluate how such changes might related to brain function.

**Decision letter after peer review:**

Thank you for submitting your article "Reduced purine biosynthesis in humans after their divergence from Neandertals" for consideration by *eLife*. Your article has been reviewed by 3 peer reviewers, including Genevieve Konopka as the Reviewing Editor, and the evaluation has been overseen by Diethard Tautz as the Senior Editor.

The reviewers have discussed the reviews with one another and the Reviewing Editor has drafted this decision to help you prepare a revised submission.

This study is one of the first to investigate how human-specific coding changes have a functional impact, in this case upon amino acid metabolism. The data support the idea that the A429V change in modern humans has played a part in brain function, notably with regards to purine biosynthesis.

Summary:

This manuscript explores the functional consequence of amino acid substitutions in modern humans compared to archaic humans. It is likely the first manuscript to functionally study one of the few such changes that have been noted among these species. The authors investigate the metabolomes of human and non-human primate tissues, noting prominent differences in purine biosynthesis in brain regions. They focus on an enzyme, adenylosuccinate lyase (ADSL), which is one of about 100 proteins which have sustained human-specific changes in their coding sequences. The functional impact of this human-specific mutation is interrogated by making a humanized mouse model which reflects reduced ADSL activity in tissues as well as reduced purine biosynthesis in brain regions. The authors also assess biochemistry of this mutation, finding it impairs protein stability, which confirms prior findings. The authors finally show that this amino acid in human iPSCs, relative to those carrying an ancestral mutation or to chimpanzee cells, confers a reduction in purine biosynthesis. Overall, the amount of experimental work is impressive and makes an important contribution to understanding molecular evolution. However, the manuscript could be improved by either providing additional support for some of the claims or revising the text, and the statistical analyses require a good deal of attention.

Essential revisions:

1. The introduction and the discussion need to discuss how this study builds on or otherwise informs previous work. Are these results unexpected based on the literature? What were the benefits of these extensive experiments over previous work?

2. Human iPSCs (409B2 cells) are used to assess the function of the A429V mutation by comparing metabolites between human cells, cells mutated to have the ancestral sequence, and chimpanzee cells. However, these experiments are only done in iPSCs, and this is a missed opportunity to evaluate species differences in cells relevant for the brain. This is a surprising gap in the story, given that significant metabolic differences between human and other primates, as well as in humanized mice (p. 7), are mainly significant in the brain. The authors should address in the discussion how using a non-neuronal cell line is informative, in light of their finding that the metabolites are highly enriched in the brain. Alternatively, the authors could relax the emphasis that these results are brain specific. Also, no mention of the chimpanzee cells occurs in the supplemental document, so this needs to be addressed and clarified.

3. The text highlights differences in oxidative phosphorylation throughout, but the evidence for changes in this pathway are not included in any of the main text figures or statistics provided in the text. This conclusion also does not seem to be supported by the supplemental tables either. It is unclear why oxidative phosphorylation is mentioned.

4. The statistical analyses chosen need to be better explained in the text and in the figures. In figures 1, 3, and 7, please provide the statistical test and cut-off in the figure legend that determined significance. Perhaps sizes of pie charts, or numbers could be added to make these more informative. As shown, these figures make a case that many metabolites of the same pathway are affected but not the extent to which they are species divergent. The authors report concordant findings in metabolomics between human-primates and the humanized mouse models. But they only report directionality, and it is unclear how the fold changes observed in activity in humanized mice compare to metabolite measurements between human and primates and also in specific brain regions? This measure may reinforce the value of the humanized model. In figures 2 and 8, it is unclear what the dots and error bars represent and which statistical tests were used. In Figure 2, only a single p value is included to describe results in the text, but they should also perform post-hoc analyses to assess significant differences within tissues. In figure 4 it is also unclear what the error bars indicate and there are no error bars in panel 4B and 6A. It is unclear to what extent these assays reflect multiple experiments, which is needed to be rigorous. In the supplemental methods, the metabolite differences were determined by a t-test – are they normally distributed?

5. In the discussion the authors indicate that it is possible the A429V substitution in ADSL has contributed to human-specific differences in brain development and function. They go to the effort of making humanized mouse models but only report there are no overt phenotypic differences. To what extent have they examined the brains of these mice? A full behavioral study is beyond this paper however some insights and discussion (of even the negative data) here would be useful.

6. In figure 8 the authors assess phosphorylation levels of AMPKa in both ancestralized and chimpanzee cells. The value that is most meaningful here is normalized to total levels, which doesn't show significant change in the ancestral sequence. Further the relative differences are smaller than that seen comparing human and chimpanzee cells. The value of these data is questionable, as there are many factors which could activate AMPK, that may be present in the chimpanzee cells.

---

## [Author Response]

Essential revisions:1. The introduction and the discussion need to discuss how this study builds on or otherwise informs previous work. Are these results unexpected based on the literature? What were the benefits of these extensive experiments over previous work?

The Introduction now discusses the previous work on BOLA2 (Nuttle et al.) and NOVA1 (Trujilo et al.) in addition to previous work on ADSL (Van Laer et al., 2018). The Discussion briefly reviews previous work on differences between humans and apes and stresses that this is the first example of a difference where a single nucleotide change results in a difference observed in the mature organism.

2. Human iPSCs (409B2 cells) are used to assess the function of the A429V mutation by comparing metabolites between human cells, cells mutated to have the ancestral sequence, and chimpanzee cells. However, these experiments are only done in iPSCs, and this is a missed opportunity to evaluate species differences in cells relevant for the brain. This is a surprising gap in the story, given that significant metabolic differences between human and other primates, as well as in humanized mice (p. 7), are mainly significant in the brain. The authors should address in the discussion how using a non-neuronal cell line is informative, in light of their finding that the metabolites are highly enriched in the brain. Alternatively, the authors could relax the emphasis that these results are brain specific.

The reduced levels of purine metabolites are seen in all tissues analyzed (Supplementary Table 5), but is most pronounced in the brain. We now point this out more clearly (page 6, first paragraph).

Given that the difference occurs in all tissues, we have refrained from differentiating the iPSCs to neurons. This would be a substantial effort particularly with regard to get different cultures to differentiate to the same extant and produce homogeneous cell types (this is something we struggle with in other projects). Thus, as the differentiation would likely just result in a somewhat larger effect, we do not think that the experimental work needed would be justified.

Also, no mention of the chimpanzee cells occurs in the supplemental document, so this needs to be addressed and clarified.

Thanks for pointing this out. The chimpanzee cell lines (and the newly added macaque cell line) are now described in the second paragraph of the Materials and methods section.

3. The text highlights differences in oxidative phosphorylation throughout, but the evidence for changes in this pathway are not included in any of the main text figures or statistics provided in the text. This conclusion also does not seem to be supported by the supplemental tables either. It is unclear why oxidative phosphorylation is mentioned.

Oxidative phosphorylation is lower in humans in the three brain regions analyzed but not in muscle and kidney (Suppl. Table 5). We do not focus on it in this paper but find it worthwhile to point out. We just mention it in the Results (page 5) and once in passing in the Discussion together with the more numerous differences in amino acid metabolism.

4. The statistical analyses chosen need to be better explained in the text and in the figures. In figures 1, 3, and 7, please provide the statistical test and cut-off in the figure legend that determined significance. Perhaps sizes of pie charts, or numbers could be added to make these more informative. As shown, these figures make a case that many metabolites of the same pathway are affected but not the extent to which they are species divergent. The authors report concordant findings in metabolomics between human-primates and the humanized mouse models. But they only report directionality, and it is unclear how the fold changes observed in activity in humanized mice compare to metabolite measurements between human and primates and also in specific brain regions? This measure may reinforce the value of the humanized model.

The new figures 2, 5 (replaces previous Figure 3), and 9 (replaces previous figure 7) now show analyses where the fold-changes purine metabolites are compared to all other metabolites detected. The new Supplementary Tables 1-3 give the fold-changes of all metabolites that differ significantly between humans and either chimpanzee or macaques in prefrontal cortex or cerebellum.

For technical and logistic reasons, the measurements of the metabolomes of the tissues from primates, from the mice, and from the cells were done on different technical platforms and at different times. Therefore, the magnitude of the differences cannot be compared between the different sets of experiments. For the cellular experiments (Figure 9), the edited cells were measured together with human and chimpanzee cells so that quantitative comparisons are possible. In that case, the increase in purine metabolites in the cells carrying the ancestral ADSL are if similar magnitude as the difference between chimpanzee and human cells, suggesting that substitution in ADSL explains much or even all of the difference observed between the species.

In figures 2 and 8, it is unclear what the dots and error bars represent and which statistical tests were used. In Figure 2, only a single p value is included to describe results in the text, but they should also perform post-hoc analyses to assess significant differences within tissues.

In Figure 4 (previous Figure 2) bars in the box plots represent the range of the data and dots outlier observations. This is now explained in the figure legend. Similarly in Figure 9 (previously Figure 8).

The “single” p-value in the text refers to that all tissues and ages are significant. This is now clarified in the text (page 7).

In figure 4 it is also unclear what the error bars indicate and there are no error bars in panel 4B and 6A. It is unclear to what extent these assays reflect multiple experiments, which is needed to be rigorous.

In Figure 6 (previous Figure 4), the legend now explains the errors bars in panel a. The legend also clarifies that these represent three experiments using two batches of proteins. The legend also clarifies that the experiments in panel b represents the average of four experiments.

Similarly, the legend to Figure 8b (previous Figure 6b) now clarifies the number of experiments summarized by the graph.

In the supplemental methods, the metabolite differences were determined by a t-test – are they normally distributed?

To test whether metabolite intensity data are normally distributed we used the Shapiro test. The null hypothesis is that the data are normally distributed, so if the p-value is greater than 0.05, then the null hypothesis is not rejected. For the majority of metabolites the p-values are greater than 0.05 (Author response image 1). After multiple test correction, the adjusted p-values are greater than 0.05 for all metabolites except three (Author response image 1). In general, the distribution of metabolite abundances is close to normal (Author response image 1). Thus, we assume that the t-test can be applied in this case.

**Author response image 1. respfig1:** Distribution of metabolite abundances. (A) Shapiro test p-values calculated for each metabolite. (B) BH-adjusted Shapiro test p-values. (C) Distribution of log2-transformed metabolite abundances (mouse liver as a typical example).

5. In the discussion the authors indicate that it is possible the A429V substitution in ADSL has contributed to human-specific differences in brain development and function. They go to the effort of making humanized mouse models but only report there are no overt phenotypic differences. To what extent have they examined the brains of these mice? A full behavioral study is beyond this paper however some insights and discussion (of even the negative data) here would be useful.

We observe no obvious morphological or behavioral phenotypes in the mice. It would require detailed and careful work to look into this in a rigorous way. Such work will be initiated in collaboration with others. In the last sentences of the Discussion we now just say that no overt phenotypes are seen.

6. In figure 8 the authors assess phosphorylation levels of AMPKa in both ancestralized and chimpanzee cells. The value that is most meaningful here is normalized to total levels, which doesn't show significant change in the ancestral sequence. Further the relative differences are smaller than that seen comparing human and chimpanzee cells. The value of these data is questionable, as there are many factors which could activate AMPK, that may be present in the chimpanzee cells.

We agree that this may requires further work and have removed the AMPK analyses from the paper.